# Sustainable Distance Online Educational Process for Dental Students during COVID-19 Pandemic

**DOI:** 10.3390/ijerph19159470

**Published:** 2022-08-02

**Authors:** Maria Antoniadou, Christos Rahiotis, Afrodite Kakaboura

**Affiliations:** Department of Operative Dentistry, National and Kapodistrian University of Athens, 11527 Athens, Greece; craxioti@dent.uoa.gr (C.R.); akakabou@dent.uoa.gr (A.K.)

**Keywords:** academic performance, COVID-19, dental education, distance learning, e-learning, e-exams

## Abstract

In this study, we evaluated the perception of distance online learning in undergraduate dental students in two different European countries during the second lockdown of the COVID-19 pandemic to explore sustainable undergraduate educational and examination e-learning forms. Dental students from Dental school of Athens, National and Kapodistrian university of Athens (N1_3rd preclinical year = 131, N2_4th clinical year = 119) and Dental school of Copenhagen (3rd preclinical year N3 = 85) completed the mixed-designed Dental e-Learning process Questionnaire (DeLQ) distributed in a google form. Responses to closed-ended questions were collected on a five-point Likert scale. Descriptive statistics were applied, and non-parametric Kruskal–Wallis tests were used to examine student groups. N1 (90% strongly agree) students reported that “e-learning is a suitable education method for theory in dentistry” at a significant level and more often than N2 (43% strongly disagree). N1 and N2 students strongly agreed that they preferred face-to-face teaching rather than distance e-learning. A relatively low number of N1 (31%) students believed that e-learning prepares them sufficiently for their practical training while none of the (0%) N2 cohort agreed. A low percentage of students in both years (N1 = 31%, N2 = 23%) believed that e-learning prepared them for their exams. Additionally, N1 = 60% and N2 = 66% preferred hybrid learning. Only 26% (N1) and 19.5% (N2) desired e-learning to continue after the COVID-19 pandemic. Nearly half of the participants believed the online exam model to be unreliable (N1 = 49%, N2 = 43%). Overall, students considered distance e-learning as an educational method applicable only to theoretical lessons. However, the lack of physical communication and interaction in distance learning led students to prefer a blended method. Students of the two faculties seemed to agree on many points, but there were also specific differences attributable to the differences in the programs and educational culture of the two countries.

## 1. Introduction

The coronavirus disease, known as the COVID-19 pandemic, has affected human activities and the healthcare systems worldwide drastically, including dental care practice [1] and education in dental academic fields [2,3,4,5]. Furthermore, subsequent enclosures due to the pandemic necessitated a direct shift from traditional in-person education to distance online learning at all academic levels [6] in health sciences [7], including dentistry [8,9].

Online teaching and learning were employed in academic environments and especially in medical sciences long before the COVID-19 pandemic led to the necessity for changes [10,11,12]. In dentistry, however, online education was not applicable because practical skills require laboratory and clinical exercises in conjunction with theoretical knowledge [13,14,15,16]. However, the consequences of the coronavirus forced dental schools worldwide—in Europe [9,17], Asia [18], the Middle East [19] and Saudi Arabia [20], Africa [21], Brazil [22], the United States [23], and Canada [24]—to disseminate theoretical knowledge exclusively through distance learning models using several digital tools [16,25].

As some of the changes in curricula across the pandemic era may positively contribute to the learning process, it is critical to record students’ views on the state of dental education and on the issues that need to be improved to accelerate the digital transformation of the traditional dental curricula and move towards a more sustainable dental educational model. So far, an incredible amount of research and review documentation has been published concerning the impact of distance e-teaching methods on dental education [26]. Methodologically, the relevant research was performed by distributing questionnaires to academic personnel [27,28], undergraduate [17], and postgraduate students of different semesters [18,24,29,30] or by conducting reviews [8,31,32].

According to the educators, one of the main issues derived from the sudden need for e-learning changes in dental education included problems in teaching and the quality of learning. These issues include complications in training on practical skills [25,33,34], concerns regarding the quality assurance of dental education [23], online practical considerations [22,29,30,35], learning challenges due to social distance, negative emotions, and other limitations [36], the amount of content shared in e-learning classes and platforms [21,37] as well as the examination process [38]. Students’ main electronic distance education concerns include difficulties in preparing for online education [9,20], the criticism that most of the online courses are in the form of lectures with only brief practical, interactive training content [30], the absence of interaction [12] and lack of motivation [39].

Although dental students seemed flexible in adapting to the digital e-learning or blended teaching environment [19,20] and were willing to accept it as a source of sharing theoretical knowledge, significant differences were detected among the dental schools and between individual contributors [9,35,40,41]. Technical problems, knowledge, use of electronic equipment, socializing and personal parameters of the teaching process, the context and structure of the online courses, previous experience, and other reasons seemed to play a significant role in the diversity of the opinions recorded [40,41]. This issue suggests that each dental institute’s e-learning acceptability and relative parameters should be investigated individually because of the differences in dental curricula and educational culture. At the same time, possible comparisons among different digital curricula may offer practical conclusions for implementing international standards in digital dental education. Within this scope, the perception of distance-learning and re-examination processes was investigated for undergraduate dental students in two different European countries during the second lockdown of the COVID-19 pandemic (April 2021). In addition, a comparison of undergraduate students’ attitudes, preferences, the effect of the academic semester on their studies, their e-background from home, other demographic factors, and their estimation of preparedness for e-exams were examined and analyzed to produce a sustainable educational model for future emergency educational use.

## 2. Materials and Methods

The study protocol was reviewed and approved by the Ethical and Research Committee of the Dental School of Athens, National and Kapodistrian University of Athens (school A) (Ref # 467). At the end of the academic year, 2020–2021, all third-year (preclinical) (N1 = 131) and fourth-year (clinical) (N2 = 119) dental students of a European school) were asked to complete a questionnaire. The same questionnaire was delivered with the same methodology applied to the third-year (preclinical) students (N3 = 85) in Dental School BDental School of Copenhagen, University of Copenhagen.

The Dental e-Learning Questionnaire (DeLQ) (Appendix A) consisted of three sections, and it was of a mixed design mode, integrating data through a Likert-scale. The first section contained four questions related to the participants’ general characteristics. The second section investigated the students’ attitudes regarding the distance e-learning implemented for all curriculum courses. In addition, 18 closed-ended statement questions were arranged with a five-point Likert rating (1: Strongly disagree, 2: Disagree, 3: Neutral, 4: Agree, and 5: Strongly agree). The third section included 3 open-ended questions related to the opinion of the students on the advantages and disadvantages of the distance e-learning education model and suggestions for improving the method.

The construct validity of this questionnaire was confirmed by a panel of 3 professors, all of whom were experts on dental education in each school, who reviewed and revised the survey questions to ensure they were relevant to the topic and expressed correctly in each language. In addition, the instrument was pilot tested on 10 students in each school to confirm its comprehensibility and usability.

The teaching structure of the courses before and during the COVID-19 pandemic era is shown in Table 1.

Only the curriculum courses’ theory was taught during the pandemic with an exclusively distance e-learning model regarding schools A and B. At the research time, in-person contact was not allowed in both schools. A variety of tools were applied for the delivery of education. All courses used the virtual classroom method through Google Meet, Zoom, and Webex video conference platforms. Students attended via electronic devices according to the timeframe of each course in the pre-pandemic curriculum. In addition, digitized educational material was uploaded in an asynchronous e-platform supported by the institutes, such as slideshows, videos, lecture recordings, relevant websites, and literature. Electronic formative and summative tests were conducted for all courses in terms of exams.

The questionnaire was developed using Google forms. The link to the questionnaire and consent form was e-mailed to each one of the participants in April 2021, with details about its completion described in the introduction of the online form. Participation was entirely voluntary, and all questions had to be completed for the form to be submitted. Every student at both study schools had the same opportunity to participate and fill in the questionnaire. No personal data or other identifiers were collected by completing the form, and the software guaranteed anonymous submission. The message to complete the survey was e-mailed twice to all participants within one week. After this period, no late responses were collected, and the questionnaire was closed to new answers. No compensation of any kind was offered to participants for participating in the study. Data were then collected and analyzed by the same researcher.

For each closed-ended Likert scale question, mean scores were calculated by gender and studies. Descriptive statistics were applied, and non-parametric Kruskal–Wallis tests were used to examine differences between the student groups. Cronbach’s alpha index was used for the reliability calculations. All reported probability values (*p* values) were compared with a significance level of 5% (*p* < 0.05). The analysis of coded data was performed using the IBM SPSS Statistics for Windows, Version 26.0 (SPSS Inc., Chicago, IL, USA).

## 3. Results

Cronbach’s alpha reliability for the questionnaire was 0.726 for school A and 0.746 for school B. Regarding school A, 148 participants from the two study years replied to the online questionnaire, yielding a response rate of 59.2%. The study sample was composed of 97 third-year (74% response rate) and 51 fourth-year (42.8% response rate) dental students. Of the participants, 96 (64.8%) were females and 52 (35.2%) males. Overall, 100% percent of the families use a computer, and all the parents of the students had obtained a university degree. Thirty-nine (*n* = 39) students from school B completed the questionnaire, 23 (59%) were females, 16 (41%) were males, and the total response rate was 46%. Overall, 100% percent of the families use a computer, 88.3% of the parents had obtained a university degree and 11.7% a high-school degree. The results of the students’ responses regarding their attitudes and experience with distance e-learning education in school A are shown in Table 2.

In both study years, half of the students agreed or strongly agreed that the e-learning education was well organized and structured, with no significant difference between the third- and fourth-year students. At the same time, nearly 30% recorded neutral opinions (neither disagree/neither agree). In addition, almost half of the third- and fourth-year students agreed that they could be prepared earlier for each lesson compared to face-to-face courses in the past.

Although the difference was not statistically significant, a higher number of third-year students were not concerned with the occurrence of technical problems during the virtual classroom presentations. Additionally, there was a significant difference in responses between third- and fourth-year students about the statement “e-learning is a suitable education method for theory in dentistry”; as 90% of third-year students strongly agreed, whereas 43% of fourth-year students strongly disagreed.

Students in both years strongly agree that they preferred face-to-face teaching over than distance e-learning. However, a relatively low number of third-year students (31%) believed (strongly agree and agree) that e-learning prepares them sufficiently for their practical training (lab exercise), and none (0%) of the fourth-year students agreed with this statement (clinical practice). A high percentage of fourth-year students held a strong negative opinion on this particular question (71% strongly disagree or disagree).

A higher percentage of third (64.5%) than fourth-year (34.5%) students stated that they ask questions to the educator and make comments in the same way as if they were attending lessons physically, although differences were insignificant. In addition, the same profile in answers was recorded when the students were asked if they often addressed questions online compared to the face-to-face lessons.

In addition, the students in both years were satisfied with the teachers’ answers in the virtual presentations (third- 67%, fourth-year 81%, strongly agree and agree). A low percentage of students in both years (31% and 23% in third- and fourth-year students, accordingly) believed that e-learning prepared them adequately enough for their exams. Additionally, 60% and 66% of the third- and fourth-year responders answered that they prefer blended learning. Almost 50% of the students in both study years reported quick adaptation to distance e-learning education.

A very high percentage of the students missed lessons in the classroom and personal communication with the teachers (78% third- and 82% fourth-year students). A relatively low number of students (24.5% third- and 29% fourth-year students), agreed and strongly agreed that e-learning is a waste of time for health science education. Only 26% (third year) and 19.5% (fourth year) of the students’ expressed a desire for distance e-learning to continue after the COVID-19 pandemic. Almost half of the students (52% third and 48% fourth year) felt uncomfortable having their web camera on when asking questions and when the teacher addressed them (48.5% third and 47.5% fourth year).

The fourth-year students were equally divided (43%) between the options that distance online exams are or are not as reliable and fair as exams conducted in a classroom. On the contrary, when third-year students compared these two exam methods, half (49%) replied that the distance online model is unreliable and unfair.

Concerning the difference between genders, a statistically significant difference was detected. A higher number of males agreed and strongly agreed with this statement than females.

The responses related to the distance e-learning questions of the third-year students from school B are presented in Table 3 and compared with the corresponding ones received from school A.

Statistically significant differences between the dental schools were detected in responses to the following questions. Fewer students studying in school B considered distance e-learning as a suitable education method for theory in dentistry (32%-B vs. 94%-A strongly agree and agree). In contrast, more students believed it can prepare them well for their practice (50%-B vs. 31%-A strongly agree and agree). In addition, a lower number of students from school B stated that they asked the teacher questions in the virtual classroom relative to what during in-person lessons (47%-B vs. 64.5%-A strongly agree and agree). At the same time, a higher percentage of those students characterized e-learning as a waste of time for education in health sciences (53.5%-B vs. 24.5%-A strongly agree and agree). On the other hand, more students in school B reported that they agreed to continue the distance e-learning education even after the COVID-19 pandemic (47.5%-B vs. 26.5%-A strongly agree and agree).

A total of 120 out of 148 students in school A and 29 out of 39 in school B commented on the three open-ended questions. Regarding school A, the ranking of advantages mentioned was “saving time” (31.6%), “avoiding travel” (19.2%), “better participation and concentration in the course due to the technological means available” (15.8%), and the “comfort of attendance” (10.8%). The corresponding advantages underlined in school B were “comfort of attendance” (60%), “saving time” (40%), and the “ability to record the lectures and watch again” (35%). 

The disadvantages identified by students in school A were “lack of personal contact and the impersonality of the process between students and professors” (32.5%), “lack of direct contact with the subject, monotony, boredom” (11.6%), “inability to respond to laboratory courses” (10.8%), “lack of interactivity and transmissibility of professors” (12.5%). On the other hand, according to the responses in school B, “lack of interaction” (60%), “difficulties concentrating and focus” (30%), and the “technical problems” (5%) were the main disadvantages.

Students’ proposals in school A were summarized as follows: “to improve the distance online teaching model “ (15.8%), “greater interactivity (with videos, multiple-choice questions, better slides, encouragement to participate with questions, week homework, day questionnaires, and available cameras), “e-learning not to continue after the pandemic since it is ineffective and destroys their relationships with fellow students and professors” (14.2%), “incorporation of e-learning only in theoretical subjects” (4.2%). Students at school B suggested learning to be continued for the theoretical lectures” (45%), “greater interactivity” (36%) and “audience to be split into smaller groups” (13%)

## 4. Discussion

This paper intended to study dental students’ perception of distance online teaching during the second COVID-19 lockdown period (April 2021). It is well reported that during this pandemic, dental education required several adaptations, with e-learning frequently utilized worldwide [9,42]. The specific time chosen to release the DeLQ questionnaire provided critical data in our study. First, it covered the experience of two semesters with distance online education. In that period, no preclinical or clinical exercise had taken place. Participants were also exposed to the distance examination experience as mentioned elsewhere [18,21,39]. Finally, it was the second year of the pandemic, allowing us to gather more knowledge about the disease and the e-learning system. Consequently, the fear, anxiety, and stress had subsided and did not dominate students’ psychology. This could especially have been the case for school A since the country was one of the least affected by the disease during the study period [43]. The third-year students were selected for schools A and B because they represent the preclinical practice years that prepare students for the clinical subjects. Additionally, the fourth year of school A is the first year in which clinical exercises begin. Accordingly, we wanted to measure the effect of the second lockdown for academic years in which preclinical and clinical practice experience would take place. We excluded the fourth-year students from school B because the institute leader ensured the continuation of practical activities during the second lockdown. Therefore, the clinical education continued unchanged for a few periods at school B.

Adaptation to educational changes was the first factor to consider in our study. Both schools in our study responded satisfactorily to educational changes, at least with regard to the high percentage of students who responded that distance online education was well organized (50%). The latter is mainly attributable to the preparedness of educators who offered educational material in digital form even before the onset of the pandemic. For them, it seemed easy to upload or/and update this material on e-platforms and transition from the face-to-face structure to e-learning in a primary stage, as mentioned elsewhere, even before the pandemic [44]. On the other hand, as reported [8], well-organized distance learning leads to the quick adaptation of students to the distance e-learning. The study revealed that up to half of school A students and more in school B had a quick reaction to the reported changes, as also mentioned elsewhere [45]. It seems that adaptation to distance education depends on the assimilation and familiarization of new technologies by both the instructor and the student [46]. Nowadays, students belong to the second-millennium generation and are familiar with digital environments [46]. This feature was evident since the families of all participants in our study used a computer. In addition, students in both dental schools possessed at least basic knowledge and skills for using the e-learning platform, which was already in function for delivering educational material and other related activities the years before pandemic. Thus, e-learning was not a severe barrier for them, as recorded in other investigations [47] and during the pandemic [48].

The time needed for adaptation to educational changes was another critical issue. No difference was detected between the preclinical and clinical academic years of students at school A in terms of how quickly they adjusted to the implemented e-learning system. This finding contradicts previous studies that reported conflicting results with senior dental students adapting more readily [48,49] than students in their early academic years [29]. It is noticeable that technological equipment and internet connection quality, which are frequently recorded as barriers in distance e-learning, did not seem to raise difficulties in the current study [38]. It is probably the main reason for the quick adaptation of the students to the fully online curriculum.

Additionally, diminished interaction in online courses was reported in our study as also mentioned elsewhere [50]. Students at school B seemed to ask questions significantly less frequently than the same year students at school A. The difficulty in focusing during the lesson and some of the technical troubles mentioned by the students at school B are elements that were not pointed out by those in school A, and possibly justified their reduced interest in addressing questions. Technical problems can also be a source of annoyance and discouragement, as mentioned in other studies [51]. In addition, engaging enough senior students at school A who had less interest in asking questions was difficult because their perception was that online teaching does not adequately support theory learning and preparation for clinical practice [50]. Concerning the instructors’ ability to answer students’ questions during the virtual classroom, this was not affected by the teaching method implemented. The students seemed satisfied with their answers in any case, as also reported elsewhere [51].

The absence of physical contact is another issue for discussion. Our results showed that students did not prefer to open their cameras during lessons, thus minimizing the physical presence in the e-classroom. This is possibly due to connecting from private places and considering the available camera to be a violation of their privacy, as discussed in other studies [36,52,53]. However, keeping the cameras off, participants’ facial expressions—which is one of the most critical elements of human communication [54]—was missed, resulting in less satisfaction from the educational process [55]. In our findings, a high percentage of students at both dental schools stated that the physical classroom lacks something, which was also discussed in the study of Wang et al. [40]. Additionally, in another study performed in 42 colleges and universities in China, it was reported that the “digital interaction between teachers and students” received the lowest satisfaction rate [41]. The latter suggests that human contact and communication are not complementary to studies [54]. Although not statistically significant, the percentage of keeping the camera off was lower among the students at school B. This probably has to do with the social and national culture and the attitude of the society of each country and the reluctance to participate in a familiar environment at a professional level, as also discussed elsewhere [56].

Additionally, time savings provided by distance e-learning allowed students to prepare before attending each course and was identified as a significant advantage in our study. This was confirmed by many students (40%) as a vital advantage of this teaching system [57]. Due to the obligatory lockdowns, students’ spare time was augmented mainly due to the absence of transportation to their school. Time management is crucial also for all health professionals and educators who can use the additional time to enrich their digital material and/or encourage students to self-study and engage in more active learning [17,57].

The impact of e-learning on the effectiveness of the exam system is another critical issue of discussion. Remote exams were conducted from a distance and considered a primary assessment model for students’ academic progress during the pandemic. Remote exams were well documented as approved by many educational institutes, years before the pandemic [58,59,60]. In both dental schools, one-third of the students felt they gained the essential knowledge to proceed to exams after fully distanced online education. The latter may be explained by the comfort of time; they were at home for a more extended period, had more time available to study, and did not waste time travelling to schools. We should point out that although most of the students stated a preference for face-to-face teaching and mentioned disadvantages such as the lack of interaction, these do not seem to be considered barriers to reaching the learning objectives of the courses. Of course, we should consider that the specific opinion of the students cannot be translated as an actual impact of the online model on learning outcomes because these are self-reported data. Nevertheless, it is noteworthy that only one-third of the participating students perceived that they were not ready to sit exams as this number is significantly lower than those reported in other relative surveys [59,60]. The lack of laboratory and clinical practice probably increased the feeling in this group of students that they were not adequately prepared for exams.

In the current study, nearly half of fourth-year students in school A and third-year students from school B considered e-exams to be reliable and fair. In addition, a significant number of the students from the same groups had a neutral opinion. Under the consideration that it was the first time that students were asked to sit electronic exams on such a large scale, under unknown conditions in terms of their structure and model, the remote exams seemed to be well accepted. On the contrary, third-year students in school A were seriously concerned about the reliability and fairness of the distance e-exams. Hence, we concluded that a significant pool of students viewed this process negatively. On the other hand, our study revealed that a significant percentage of students were neutral. So, a significant portion of students had positive or neutral views on the issue. Within this scope, if a dental school wanted to integrate distance e-exams into normal operation conditions, it could do so by adopting improvements to the system, including proctoring solutions, modifications of the exam structure, and by considering the grade as a compulsory pass/fail rather than regular grading systems as mentioned elsewhere [61].

Interestingly enough, gender differences were not seen to affect our study as shown in the results from both schools. A previous study about distance and in-person learning showed a difference in the response across genders, where females preferred face-to-face learning at a higher percentage than in our study [57]. It is difficult to explain this tendency considering that before the pandemic, there was a significant effect of gender, computer use, and self-perceived computer experience in terms of the attitudes of computer anxiety in students. Previously, males were found to have less computer anxiety than females [62]. Furthermore, students who have used computers for a more extended period and those with a higher self-perception of experience also showed less computer anxiety. Finally, it was mentioned that the influence of computer experience works in different ways for males and females. Computer experience has finally positively impacted a decrease in the computer anxiety for men, but a similar effect was not found for women [62]. It is probable that gender psychology, cultural and national attitudes, and beliefs may explain the differences in responses among the relevant studies.

Overall, we reported that differences exist in the perception of the effectiveness of the e-learning system in the two European countries. A high proportion of the students in both academic years and dental schools preferred the face-to-face learning. The very high percentage of responses probably indicates that distance online teaching deprived students of personal communication/contact in the classroom with their fellow students and teachers which consequently affected their psychology. Additionally, interactivity is something they desired to a greater extent. The modification of the teaching model makes the lesson more enjoyable, less monotonous, and includes components that activate them. New and innovative teaching concepts, especially those focusing on theoretical knowledge, were applied to e-learning platforms [36,63] and are certainly required for a sustainable future hybrid dental educational process.

However, during the pandemic, educators lacked the training required to adapt to the new teaching model or had no experience and skills to apply accordingly. Furthermore, lack of communication may be vital as it emerged to be a decisive factor in shaping this view. Dental students in our study were pessimistic about the fully online learning method. Still, they would accept the implementation of blended education in a high percentage (60% and 70%, respectively), as also discussed in other studies [17]. Acceptance of e-learning as a sustainable dental education tool can be ensured if time saving, digital interaction, and specification of the learning objectives of the courses are implemented. Those seem to be significant incentives for students to accept a blended dental curriculum [17,57]. In our study, a combination of conventional and digital teaching was identified as the best approach to achieving educational success goals and driving optimum performance as discussed elsewhere [64,65,66,67]. For online learning in the future in dental education, the adoption and use of online education programs requires a shift in the view of traditional approaches to learning to a more flexible perspective with the integration of new educational tools. When planning such programs, we must consider that the process of learning moves away from a time-based paradigm to focusing on measuring the acquisition of skills and abilities free of the traditional educational time-constraints approach [68]. This will require reviewing the learning objectives to align them with virtual environments, by considering how available technologies can facilitate course delivery in virtual environments while maintaining course objectives, and employing teaching strategies grounded in learning theory that is germane to virtual teaching environments, such as cognitive load theory, refining assessments to reflect best practices in virtual environments and maintaining alignment with learning objectives. Finally, educators should be mindful of students and maintain compassion, as they are also transitioning to virtual education environments [69].

Conclusively, our findings offer supporting evidence on the effectiveness of distance e-learning, specifically in the preclinical subjects of undergraduate dental education in two different educational systems. Further research is needed however to clarify the effects of e-learning and the conditions under which it can be more effectively implemented in theoretical or clinical subjects for a sustainable universal dental curriculum.

Our study cannot be considered as an entirely objective assessment of distance learning. The experience of e-learning our students faced during the pandemic was not formed in normal living conditions. The pandemic created psychological pressure and affected students’ personal, family, and financial life. Additionally, our results should be evaluated with caution because it did not concern an online educational process provided remotely. There was no transitional period, and both students and teachers were unfamiliar with relevant technology and the different modes and tools of teaching. Additionally, in the virtual classrooms, the teaching model of the on-site classes was applied precisely because such a sharp transition did not leave time for modifications and adaptations of the educational material and the teaching methods in the distance model. For example, there was no way to replace the hours of laboratory and clinical practice contained in the curriculum with forms such as virtual demonstrations, discussions on clinical cases, discussions on videos with clinical techniques, etc., so that they could judge that method as an online model. Whether e-learning is a direct or mediated factor in improving achievement needs to be assessed in depth, and the design and delivery strategies that are more effective in each academic environment need to be identified.

## 5. Conclusions

Although there was no time to transition from conventional to distance teaching, students’ adaptation was quick and they did not face problems. Students recognized that distance e-learning is an educational method that could be applied to theoretical lessons with certain advantages. They may acquire knowledge and skills that cannot be obtained through traditional learning, which compensates for the lack of information and skills identified by the DeLQ questionnaire. However, the absence of physical communication and interaction in distance learning leads students to expect a blended method mainly for preclinical theoretical subjects. The objectives of the study to address basic international online educational standards was fulfilled in our study since students of the two faculties seemed to agree on many points. Overall, the blended teaching method could be appreciated in preclinical subjects. However, e-learning methods were not found to be able to compensate for clinical exercises or the exams system despite changes in the curricula or exams procedures. The design of the assessment instruments and curriculum types used for the transition to e-learning in dental education requires further study to address issues of the knowledge, skills and abilities of both educators and students.

## Figures and Tables

**Table 1 ijerph-19-09470-t001:** The teaching structure of all the courses in the third- and fourth-year curriculum for the two Dental Schools, before and during the COVID-19 pandemic era.

		Before COVID-19	During COVID-19
Dental School A (Athens University)	3rd year 4th year	lectures 12 h/week lab exercises 10 h/week lectures 11 h/week clinical practice 18 h/week	lectures 12 h/week (distance e-learning) lectures 11 h/week (distance e-learning)
Dental School B (Copenhagen University	3rd year	lectures 12 h/week lab exercises or clinic 20 h/week	lectures 12 h/week (distance e-learning)

**Table 2 ijerph-19-09470-t002:** Results for the questions concerning distance e-learning education of third- and fourth-year students in school A (dental School of Athens).

	Year/Gender	Strongly Disagree	Disagree	Neither Disagree /Agree	Agree	Strongly Agree	*p* Value
Q1. E-learning was well organized and structured	3rd	2%	16.5%	31%	30%	20.5%	0.606
	4th	10%	14%	28.5%	42.5%	5%	
	female *	2.5%	18.5%	37%	24%	17.5%	
	male *	4.5%	11.5%	18.5%	39.5%	26%	0.551
Q2. During the e-learning, I was able to be prepared for each lesson earlier than I was during the face-to-face courses I had attended in previous years	3rd	9%	16.5%	28%	25%	21.5%	0.747
	4th	9%	16.5%	24%	28%	23.5%	
	female	13%	16%	29%	24%	20%	
	male	2%	16%	23%	28%	31%	0.092
Q3. The quality of internet connection, images, and sound during the distance lessons was good	3rd	0%	2%	2%	11.5%	84.5%	0.942
	4th	4%	16.5%	28%	42%	9.5%	
	female	4%	14.5%	41%	32%	8%	
	male	0%	14%	18.5%	59%	8.5%	0.053
Q4. E-learning is a suitable education method for the theory in dentistry	3rd	2%	1%	3%	4%	90%	0.047
	4th	43%	10%	20%	20%	7%	
	female	20%	16%	22.5%	16%	25.5%	
	male	18.5%	21%	16%	14%	30.5%	0.471
Q5. In general, I prefer “face-to-face” rather than distance e-learning education method	3rd	10%	9%	15%	17%	49%	0.73
	4th	0%	5%	9.5%	19%	63.5%	
	female	12%	6.5%	18.5%	9%	54%	
	male	0%	7%	2%	14%	77%	* 0.006
Q6. I feel that e-learning prepares me well for my practical training	3rd	28%	32%	9%	6%	25%	* 0.032
	4th	52%	19%	29%	0%	0%	
	female	30.5%	22.5%	24%	9%	14%	
	male	32.5%	28%	25.5%	14%	0%	0.160
Q7. During the online courses, I address questions and comments, as I was doing in the “face-to-face” model	3rd	7%	19.5%	9%	5%	59.5%	0.604
	4th	28.5%	14%	33%	10%	24.5%	
	female	33%	26.5%	16%	12%	22.5%	
	male	14%	28%	30%	16%	12%	0.360
Q8. In the context of e-learning, I dare to ask questions more often than I was doing in the “face-to-face” courses	3rd	10%	17.5%	5%	4%	63.5%	0.87
	4th	33%	19%	33%	5%	10%	
	female	33%	25%	24%	12%	16%	
	male	35%	32.5%	14%	5%	23.5%	0.111
Q9. The answers of educators to my questions/queries in the e-learning were sufficient	3rd	3%	5%	25%	39%	28%	0.166
	4th	5%	0%	14%	38%	43%	
	female	4%	5%	24%	37%	30%	
	male	2%	2%	18.5%	42%	35.5%	0.555
Q10. E-learning prepared me so well as I feel ready to sit in exams	3rd	7%	24%	33%	19.5%	11.5%	0.229
	4th	24%	10%	43%	19%	4%	
	female	8%	17%	37%	13%	25%	
	male	9%	18%	32.5%	25.5%	15%	0.692
Q11. I would prefer blended learning (classroom, online learning)	3rd	7%	12%	21%	26%	34%	0.507
	4th	0%	10%	24%	28.5%	37.5%	
	female	5%	14.5%	21%	22.5%	37%	
	male	4.5%	7%	21%	28%	39.5%	0.389
Q12. I am satisfied with how quickly I have adapted to the distance e-learning model	3rd	9%	18.5%	24%	28%	20.5%	0.231
	4th	5%	14.5%	28.5%	10%	41%	
	female	8%	21%	16%	28%	27%	
	male	9%	7%	35%	18.5%	30.5%	0.744
Q13. I miss the lessons in the classroom and the personal communication with the teachers	3rd	2%	2%	8%	9%	79%	0.26
	4th	0%	4%	14%	24%	58%	
	female	4%	7%	16%	22.5%	50.5%	
	male	2%	2%	10%	16%	70%	0.134
Q14. E-learning is a waste of time for health science students	3rd	26%	35%	14.5%	15.5%	9%	0.305
	4th	14%	33%	24%	5%	24%	
	female	21%	38.5%	12%	13%	15.5%	
	male	25.5%	25.5%	21%	16%	12%	0.83
Q15. After the end of the pandemic, I want to continue with the distance e-learning method	3rd	40%	21.5%	12%	5%	21.5%	0.499
	4th	47.5%	14%	19%	5%	14.5%	
	female	37%	21%	13%	4%	25%	
	male	42%	18.5%	14%	7%	18.5%	0.53
Q16. I feel uncomfortable having my camera open when I ask questions	3rd	11%	17.5%	19.5%	17.5%	34.5%	0.495
	4th	19%	14%	19%	19%	29%	
	female	6.5%	20%	22.5%	20%	31%	
	male	21%	9%	23%	7%	40%	0.434
Q17. I feel uncomfortable having my camera open when my teacher addresses me	3rd	14.5%	14.5%	22.5%	14.5%	34%	0.357
	4th	19%	28.5%	5%	19%	28.5%	
	female	10.5%	13.5%	22.5%	16%	33.5%	
	male	21%	21%	14%	14%	30%	0.203
Q18. I consider that distance online examinations are as reliable and fair as written ones in a classroom physically	3rd	24%	25%	21%	12%	18%	0.751
	4th	33%	10%	24%	19%	24%	
	female	24%	16%	24%	20%	16%	
	male	25.5%	30%	16%	11%	27.5%	0.290

* Statistically significant value.

**Table 3 ijerph-19-09470-t003:** Results for the questions concerning distance e-learning education of third-year students in Dental School B (Copenhagen University) and comparison with the corresponding answers achieved by third-year students in Dental School A (Athens University).

	University	Strongly Disagree	Disagree	Neither Disagree/Agree	Agree	Strongly Agree	*p* Value
Q1. E-learning was well organized and structured	A	2%	16.5%	31%	30%	20.5%	0.461
	B	0%	36%	21%	26%	17%	
Q2. During the e-learning, I was able to be prepared for each lesson earlier than I was during the face-to-face courses I had attended in previous years	A	9%	16.5%	28%	25%	21.5%	0.512
	B	5%	26%	31.5%	21%	16.5%	
Q3. The quality of internet connection, images, and sound during the distance lessons was good	A	0%	2%	2%	11.5%	84.5%	0.94
	B	0%	16%	12%	51%	21%	
Q4. E-learning is a suitable education method for the theory in dentistry	A	2%	1%	3%	4%	90%	* 0.04
	B	21%	21%	16%	16%	16%	
Q5. In general, I prefer “face-to-face” rather than distance e-learning education method	A	10%	9%	15%	17%	49%	0.056
	B	21%	10%	16%	31.5%	21.5%	
Q6. I feel that e-learning prepares me well for my practical training	A	28%	32%	9%	6%	25%	* 0.011
	B	16%	8%	16%	31.5%	18.5%	
Q7. During the online courses, I address questions and comments, as I was doing in the “face-to face” model	A	7%	19.5%	9%	5%	59.5%	* 0.046
	B	16%	16%	21%	31.5%	15.5%	
Q8. In the context of e-learning, I dare to ask questions more often than I was doing in the “face-to-face” courses	A	10%	17.5%	5%	4%	63.5%	0.337
	B	16%	37%	21%	10%	16%	
Q9. The answers of educators to my questions/queries in the e-learning were sufficient	A	3%	5%	25%	39%	28%	0.38
	B	0%	10%	16%	31.5%	43.5%	
Q10. E-learning prepared me so well as I feel ready to sit in exams	A	7%	24%	33%	19.5%	11.5%	0.896
	B	26%	5%	26%	26%	17%	
Q11. I would prefer blended learning (classroom, learning-learning)	A	7%	12%	21%	26%	34%	0.53
	B	5%	0%	21%	10%	62%	
Q12. I am satisfied with how quickly I have adapted to the distance e-learning model	A	9%	18.5%	24%	28%	20.5%	0.148
	B	5%	10%	21%	26%	38%	
Q13. I miss the lessons in the classroom and the personal communication with the teachers	A	2%	2%	8%	9%	79%	0.18
	B	10%	5%	21%	21%	43%	
Q14. E-learning is a waste of time for health science students	A	26%	35%	14.5%	15.5%	9%	* 0.001
	B	10%	5%	31.5%	10%	43.5%	
Q15. After the end of the pandemic, I want to continue with the distance e-learning method	A	40%	21.5%	12%	5%	21.5%	* 0.006
	B	5%	16%	31.5%	21%	26.5%	
Q16. I feel uncomfortable having my camera open when I ask questions	A	11%	17.5%	19.5%	17.5%	34.5%	0.1
	B	10%	31.5%	26%	16%	16.5%	
Q17. I feel uncomfortable having my camera open when my teacher addresses me	A	14.5%	14.5%	22.5%	14.5%	34%	0.088
	B	16%	26%	31.5%	10%	16.5%	
Q18. I consider that online distance examinations are as reliable and fair as written ones in a classroom physically	A	24%	25%	21%	12%	18%	0.259
	B	10%	16%	31.5%	31.5%	11%	

* Statistically significant value.

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
