# Peer review of "Sustainable Distance Online Educational Process for Dental Students during COVID-19 Pandemic"

_ijerph, 2022, doi:10.3390/ijerph19159470_

Round 1

Reviewer 1 Report

The article is of interest and very good timing

I see a great value of reporting on education during the pandemic and not at least with so called practical professions

The paper is well written, in structure , language and format 

References are adequate, maybe some ref could have been added as for example from  global organizations on the theme of pivoting education

Maybe alos quality issues could have been raised and also on differences o ERT and quality planned Education 

Author Response

Comments and Suggestions for Authors

The article is of interest and very good timing. I see a great value of reporting on education during the pandemic and not at least with so called practical professions

The paper is well written, in structure , language and format

References are adequate, maybe some ref could have been added as for example from global organizations on the theme of pivoting education

Answer of the authors: we added a small comment in the discussion part (lines 393-398) adding references 72, 73.

  1. Camacho D, Legare J. Pivoting to online learning-the future of learning and work. J Competency-based Edu. 2021; 6(1).
  2. Collins B, Day R, Hamilton J, Legris K, Mawdislhey H, Walsh T, Collins B. 12 Tips for Pivoting to Teaching in a Virtual En-vironment. MedEdPublish 2020, 9:170 (https://doi.org/10.15694/mep.2020.000170.1)

Maybe also quality issues could have been raised and also on differences o ERT and quality planned Education

Answer of the authors: we added a small comment in the discussion part (lines 398-403)

Reviewer 2 Report

Apart from a few mistakes in the reference list that need to be corrected, I think the paper is well written and suitable for publication.

Author Response

Comments and Suggestions for Authors

Apart from a few mistakes in the reference list that need to be corrected, I think the paper is well written and suitable for publication.

Authors response: References were checked. Thank you for your timing in reviewing our article

Reviewer 3 Report

The topic presented is an interesting one and has the ability to attract the attention of specialists in the field. I carefully read the paper and the paper is well-written. The authors carefully discussed the issue in detail and thoughtfully provided their valuable viewpoints.  The study presents a descriptive analysis.

In the conclusions section, I recommend indicating how the objectives were met. 

Author Response

Comments and Suggestions for Authors

The topic presented is an interesting one and has the ability to attract the attention of specialists in the field. I carefully read the paper and the paper is well-written. The authors carefully discussed the issue in detail and thoughtfully provided their valuable viewpoints.  The study presents a descriptive analysis.

In the conclusions section, I recommend indicating how the objectives were met.

Authors response: conclusions section was rewritten. Please see underlined text. Thank you for your comments.